# Heterodimer Formation of the Homodimeric ABC Transporter OpuA

**DOI:** 10.3390/ijms22115912

**Published:** 2021-05-31

**Authors:** Patricia Alvarez-Sieiro, Hendrik R. Sikkema, Bert Poolman

**Affiliations:** Department of Biochemistry, Groningen Biomolecular Sciences and Biotechnology Institute, University of Groningen, Nijenborgh 4, 9747 AG Groningen, The Netherlands; p.alvarez.sieiro@rug.nl (P.A.-S.); hendriksikkema@gmail.com (H.R.S.)

**Keywords:** ABC-transporter, OpuA, homo- and heterodimeric complexes, affinity purification, mechanism of multimerization, membrane protein, nanodisc reconstitution

## Abstract

Many proteins have a multimeric structure and are composed of two or more identical subunits. While this can be advantageous for the host organism, it can be a challenge when targeting specific residues in biochemical analyses. In vitro splitting and re-dimerization to circumvent this problem is a tedious process that requires stable proteins. We present an in vivo approach to transform homodimeric proteins into apparent heterodimers, which then can be purified using two-step affinity-tag purification. This opens the door to both practical applications such as smFRET to probe the conformational dynamics of homooligomeric proteins and fundamental research into the mechanism of protein multimerization, which is largely unexplored for membrane proteins. We show that expression conditions are key for the formation of heterodimers and that the order of the differential purification and reconstitution of the protein into nanodiscs is important for a functional ABC-transporter complex.

## 1. Introduction

Most proteins exist as multimeric complexes, in bacteria often as symmetric homomers with identical subunits derived from the same gene (as is evident for e.g., ATP-binding cassette transporters), whereas in (higher) eukaryotes, complex heteromers or proteins with domains that fused and evolved independently are relatively more common [1,2]. In general, multimerization is a feature that confers advantages for the cell as described by [1,3]. First, large complexes have a smaller surface area-to-volume ratio, which increases stability and may reduce protein denaturation and promiscuous interactions. Second, complexes consisting of multiple small proteins are easier to fold than single-chain complexes of similar size. Moreover, big multimeric complexes not only increase the chance for allosteric modulation of protein activity but also the frequency of substrate encounter. In addition, homomultimeric proteins have additional advantages as error control in synthesis and more efficient use of genomic space compared to heteromultimeric or monomeric complexes, thereby saving metabolic energy [1,3,4,5,6].

However, the assembly of multimers is not as trivial, as they consist of individually translated protein chains. When do the subunits assemble? How do they find one another? The answer to these questions is diverse, as highlighted in recent reviews [7,8,9]. Among the assembly scenarios, we can distinguish post- and co-translational assembly. In the former, the proteins assemble as full subunits, in the latter, they assemble during the translation process of at least one chain, see [10] for a review. Naturally, both scenarios require the subunits to be in close proximity. The proximity or local concentration of the subunits is affected by diffusion and the location of their synthesis in the cell. In bacteria, the subunits of multimeric proteins are typically expressed from the same operon and thus formed in a given order. If the same subunit is needed more than once, it may still originate from the same operon transcript (polysomal synthesis) but will be formed at a later time [11]. Alternatively, it may be formed from another transcript elsewhere in the cell, and in that case, the subunits will have to diffuse further through the membrane or cytoplasm to find each other [12]. The successful assembly of a multimeric complex will also depend on the concentration of the components and their affinity for each other [12,13]. An additional layer of complexity comes from these factors not being the same for membrane proteins and soluble proteins, as the diffusion of membrane proteins is two to three orders of magnitude slower and occurs in a 2D space compared to faster diffusion in 3D for soluble proteins, yet the orientational restriction of proteins in the membrane offers an advantage for their assembly ([14], see also the Discussion section). Furthermore, the folding, which is dominated by hydrophobic interactions for soluble proteins, is very different in membrane proteins [7,8,15].

Studying homomultimeric proteins can be a challenge, as many tools in chemical biology and biochemistry rely on the modification of cysteine residues and or incorporated non-natural amino acids. In homomultimers, any mutation made in one subunit will be replicated in the other(s), which can be problematic if one aims to label specific sites with probes for fluorescence (e.g., single-molecule Förster Resonance Energy Transfer, smFRET), electron spin resonance (e.g., Double Electron–Electron Resonance measurements, DEER) or other probe-based assays. As an example, for smFRET, a fluorescence donor and acceptor need to be introduced at specific sites, which is often done by introducing Cys or non-natural amino acids that can be labeled with the appropriate probe [16]. However, in homodimeric proteins for example, the introduction of two mutations already introduces four sites for labeling, which significantly complicates the spectroscopy. In other cases [17], a single point mutation in one protomer may be required to answer important scientific questions. 

To circumvent the problem, one could purify the protein, separate the subunits, and subsequently mix differentially labeled subunits and reassemble the protein complex. We have previously taken this approach in the study of the ATP-binding cassette (ABC) transporter OpuA from *Lactococcus lactis* [18]. OpuA is a homodimeric membrane protein composed of two membrane subunits (OpuABC) and two nucleotide-binding subunits (OpuAA). OpuABC comprises the transmembrane domain (TMD), which is surrounded by a scaffold and connected to the extracytoplasmic substrate-binding domain (SBD). The OpuA complex in the detergent-solubilized state disassembles into two OpuABC and two OpuAA subunits when the glycerol concentration falls below 15–20% (*v/v*), but the dissociation of the complex is reversible. In this way, we were able to create apparent heterodimeric complexes in which one of the transmembrane subunits was engineered and the other one not. Similar methods have been used for e.g., the membrane proteins GltPh, VcINDY, and BetP [19,20]. However, this approach is not generally applicable, especially for proteins that are not stable in a detergent environment or for which the monomers cannot be separated. Therefore, we sought for a more generic method to specifically alter one subunit of homomultimeric protein complexes prior to assembly.

Here, we present a genetic approach to form apparent heteromers from homomultimeric proteins. We reasoned that it should be possible to form apparent heteromultimeric complexes of the OpuABC subunit by duplicating the *opuABC* gene and making constructs in which one of the subunits has a metal affinity and the other has a streptavidin tag for purification, named OpuABC-H and OpuABC-S, respectively. In vivo, the following protein complexes will form: (OpuAA)_2_–(OpuABC-H)_2_, (OpuAA)_2_–(OpuABC-S)_2_ and (OpuAA)_2_–OpuABC-H–OpuABC-S, theoretically in a ratio of 1:1:2 if the OpuABC-H and OpuABC-S subunits are formed in equal amounts and have equal probability of assembling into a functional complex (Figure 1). Differential affinity chromatography can subsequently be used to enrich for (OpuAA)_2_–OpuABC-H–OpuABC-S. 

We found that nanodisc reconstitution plays a decisive role in the successful purification and functionality of the heterodimeric complex. Nanodiscs have become an essential tool for structural and functional studies of membrane proteins. They provide a native-like phospholipid bilayer environment, allowing the stability and functionality of membrane proteins [21,22]. In this paper, we present the different strategies to form apparent heterooligomeric OpuA complexes in *L. lactis* and the methodology to purify the proteins.

## 2. Results

We first constructed a series of expression plasmids for *L. lactis* and evaluated the expression and purification of homodimeric OpuA variants with three different affinity tags (Figure 2a): His_6_-tag, StrepII-tag, and TwinStrepII-tag. Each of the tags is present at the C-terminus of the OpuABC subunit. The genes are present in an operon in the order *opuAA-opuABC*, and they are cloned under the control of the tightly regulated nisin-inducible *pNisA* promoter, using the medium-copy number vectors pNZ8048 (Cm^Res^, pSH71 origin of replication) and pIL253 (Cm^Res^, pSH71 origin of replication). The pNZ8048 vector has a so-called pSH71 rolling-circle type of replication [23], whereas the pAMβ1-derived vector pIL253 is a theta-replicating plasmid [24]. Both vectors are compatible with each other and thus can be used for the co-expression of proteins in the same host [25].

### 2.1. Verification of Activity with Different Affinity Tags

To avoid recombination with genomic *opuA* genes, the plasmids were transformed into *L. lactis* Opu401, an *opuA* deletion strain that was derived from *L. lactis* NZ9000 [26]. SDS-PAGE analysis shows the successful purification of OpuA with each of the three different affinity tags. Both subunits (OpuAA and OpuABC) are present in an approximate 1:1 ratio (Figure 2b) [18,26,27,28]. We obtained approximately 110 mg of membrane vesicles per 2 L culture for each of the three strains. However, the OpuA yield was very different depending on the affinity tag used; the His_6_-tag was the most efficient with a yield of ≈18% of purified OpuA (determined by integrating the A280 signal of product peak in the gel-filtration chromatogram; in the cases where there was no SEC run of the final product, a nanodrop was used to determine the A280 signal of the elution) per mg of total vesicle protein (determined by BCA assay), followed by TwinStrepII-tag with a ≈9% yield and StrepII-tag with a ≈2.5% yield. The ATP hydrolysis activities of homodimeric OpuA with His_6_-tag, StrepII-tag, or TwinStrepII-tag were verified in the nanodisc environment. OpuA was reconstituted in MSP1D1 nanodiscs with the lipid composition of 38 mol % DOPG (*1,2-dioleoyl-*sn*-glycero-3-phosphatidylglycerol*), 12 mol % DOPC (*1,2-dioleoyl-*sn*-glycero-3-phosphatidylcholine*) plus 50 mol % DOPE (*1,2-dioleoyl-*sn*-glycero-3-phos-phatidylethanolamine*) and a reconstitution ratio of OpuA/lipids/MSP1D1 of 1:20:2000 [27].

The size exclusion chromatography profiles of the three homodimeric OpuA complexes are very similar (Figure 2c). SDS-PAGE analysis (Figure 2d) illustrates the presence of the OpuAA (47 kDa) and OpuABC (63 kDa) subunits and the scaffold protein MSP1D1 (25 kDa). The ATPase activity with and without the substrate of OpuA, glycine betaine, was determined using a coupled enzyme assay consisting of limiting amounts of OpuA in nanodiscs and an excess of pyruvate kinase plus lactate dehydrogenase activity. The different affinity tags did not significantly influence the glycine betaine-dependent hydrolysis of ATP by OpuA (Figure 2e).

### 2.2. Heterodimer Formation

Next, we transformed plasmid pNZopuAHis in combination with either pILopuAS or pILopuASS into *L. lactis* Opu401 to obtain heterodimers with different affinity tags. Theoretically, this approach can yield three different species: His_6_-tagged homodimeric OpuA (OpuA-H), Strep-tagged homodimeric OpuA (OpuA-S; OpuA-SS), and heterodimeric OpuA containing both a His_6_ and a Strep-tagged subunit (OpuA-HS; OpuA-HSS; Figure 3a). To solely select and purify the desired heterodimeric protein, we apply a two-step affinity chromatography (summarized in Figure 3b,c).

The initial protocol to obtain the hetero OpuA-HS mutant was based on an Ni^2+^-Sepharose purification to retain all His_6_-tagged complexes and thus remove Strep-tagged homodimers (OpuA-S) followed by a Strep-tactin purification to retain the Strep-tagged complexes (heterodimers) and remove His_6_-tagged homodimers (OpuA-H) or vice versa. Finally, size exclusion chromatography was used for further purification and quality control (degree of monodispersity). However, little to no protein was obtained after the two purification steps; therefore, either the heterodimeric species did not form or it was lost in the purification process.

### 2.3. Heterologous Recombination

The pNZopuAHis and pILopuAS vectors are compatible and have different antibiotic markers, but both carry homologous *opuA* sequences (Figure 2a) that may recombine and jeopardize heterodimer formation. RecA is the major protein involved in homologous recombination and DNA repair in *L. lactis* [29,30,31]. Therefore, we constructed an *L. lactis* Opu401 *∆recA* strain (Table 1). We transformed both plasmids and applied the same purification protocol as described above, but the yield of heterodimeric OpuA was still negligible (data not shown), suggesting that RecA homologous recombination is not a main problem.

### 2.4. Optimization of Induction

The overproduction of proteins can activate stress responses, which has been shown to influence protein expression in *L. lactis* [32] and other (micro)organisms [33,34,35]. One strategy to minimize this effect is to slow down the protein production by decreasing the amount of inducer or lowering the induction temperature. To optimize the induction conditions, we performed small-scale (50 mL) induction tests. After reaching a cell density of OD_600_ = 0.5, we induced with 0.05%, 0.02%, 0.01% or 0.002% (*v/v*) of culture supernatant of the nisin A-producing strain *L. lactis* NZ9700 [36] (hereafter referred to as nisin A*), and the temperature during induction was either kept at 30 °C or lowered to 21 °C for 2, 4, or 8 h (Figure 4a). For each condition, membrane vesicles were obtained, and OpuA was purified by a single Ni^2+^-Sepharose purification step, obtaining a mixture of His_6_-tagged homodimers (OpuA-H) and heterodimers (OpuA-HS) (Figure 4b). Then, the fraction of OpuA-HS was quantified by Western blotting with monoclonal antibodies raised against the StrepII-tag. Indeed, a lower induction temperature led to higher amounts of heterodimer, albeit at the expense of cell biomass from which to purify protein.

Based on the induction test, we selected two conditions for further experiments: Growth at 30 °C, followed by (i) induction at 21 °C with 0.05 % (*v/v*) of nisin A* for 2 h and (ii) induction at 21 °C with 0.01 % (*v/v*) nisin A* for 4 h. Large-scale 2 L cultures were induced, and membrane vesicles were prepared as described in the Methods section. The decrease in the temperature during induction reduced the final membrane vesicle protein yield from 55 to 37.5 mg/L, but the amount of heterodimeric OpuA relative to the amount of membrane vesicles was increased by at least 2-fold. The purification of OpuA-HS from the best expression conditions (0.01% nisin A*; 21 °C; 4 h) was analyzed by SDS-PAGE and Western blotting (Figure 5). We also observed that the yield of heterodimeric OpuA is dependent on the order of the purification steps. Strep-tactin purification followed by Ni^2+^-Sepharose purification yielded a recovery of 0.008% (6.4 μg of heterodimer from 75 mg of vesicle protein). Ni^2+^-Sepharose purification followed by Strep-tactin purification yielded a recovery of 0.12 % (88 μg of heterodimer from 75 mg of vesicle protein). Despite the improvements in the conditions, the low efficiency and recovery yield prohibited further studies, e.g., reconstitution of heterodimeric OpuA for functional analyses.

### 2.5. TwinStrepII-Tag

Fusion proteins containing two copies of StrepII-tag, i.e., TwinStrepII-tag, have higher affinity for Strep-tactin compared to those with only a single StrepII-tag, thus allowing more efficient protein purification, as we showed for the homodimeric complex (Figure 2B). To increase the yield of heterodimeric OpuA, we switched from pILopuAS to pILopuASS, which contains the C-terminal TwinStrepII-tag sequence. After two steps of purification, the yield of heterodimeric OpuA was indeed higher and increased to approximately 0.4% recovery (80 μg of heterodimer from 19 mg of vesicle protein). The four times higher recovery yield now allowed for continuation with further experiments. Moreover, the TwinStrepII-tagged OpuA subunit (OpuABC-SS) not only allowed obtaining a higher protein yield, but this subunit also migrates differently on SDS-PAA gels and clear separation from the His_6_-tagged subunit (OpuABC-H) (see Section 2.7), allowing easy visualization of the extent of heterodimer formation on SDS-PAGE. Heterodimeric OpuA-HSS was reconstituted into nanodiscs; however, SDS-PAGE analysis of SEC chromatograms profiles showed dissociation of the nucleotide-binding domain (OpuAA) from the OpuA complex, as has been shown before [18], explaining the lack of ATP hydrolysis activity.

### 2.6. Optimization of Reconstitution

Two sequential affinity tag purifications require the protein to be stable in the detergent solubilized state for up to 12 h, which is problematic for OpuA, as the complex readily dissociates, especially in low glycerol concentrations [18]. Therefore, we proceeded by performing the reconstitution in between the two purification steps. Thus, the OpuA complexes were purified by metal-affinity chromatography and then immediately incorporated into MSP1D1-based nanodiscs, which yields a population of homodimeric OpuA-H and heterodimeric OpuA-HSS nanodiscs. Within the membrane environment of the nanodiscs, OpuA is much more stable, and the glycerol concentration can be lowered to 4%, which increased the Strep-tactin purification efficiency. Furthermore, to improve the quality of the OpuA nanodiscs, we varied the lipid/protein stoichiometry during the self-assembly process (Figure 6a). The fully assembled OpuA nanodiscs, with a diameter of roughly 10 nm, elute between 9 and 10 mL. The void peak around 8 mL contains liposome-like structures, and the peaks at higher elution volumes (>11 mL) are empty nanodiscs or belt protein structures, as determined previously [27,37]. We found that a combination of higher concentration of OpuA (4.32 μM) and a ratio of OpuA/MSP1D1/lipids of 1:20:1000 yields a more separated peak fraction during size exclusion chromatography and a higher ATPase activity (Figure 6b).

### 2.7. Purification of the OpuA-HSS Heterodimer

Cells were induced under optimal condition (growth and induction temperature of 21 °C, 4 h of induction with 0.01% nisin A*), and 24 mg of membrane vesicles were subjected to Ni^2+^-Sepharose purification, yielding a total of 3.8 mg of His_6_-tagged homodimers and heterodimers (OpuA-H and OpuA-HS). This mixture was reconstituted into nanodiscs and purified by SEC. The chromatogram showed a well-separated peak fraction (Figure 7a), and the SDS-PAGE analysis showed a comparable intensity for the OpuAA and OpuABC subunits (Figure 7b). Next, the selected peak fraction was purified on a Strep-tactin resin, yielding a total of 50 μg of heterodimeric OpuA in nanodiscs, which corresponds to a recovery of 0.2%. Functionality of the heterodimeric OpuA-HSS protein was demonstrated by ATPase activity measurements (Figure 7c). We attribute the somewhat lower activity of the heterodimer, compared to similar amounts of homodimeric OpuA nanodiscs, to the loss of a fraction of the OpuAA subunit, as can be seen on SDS-PAGE gels (Figure 7b).

## 3. Discussion

One of the crucial steps in the in vivo formation of heterodimers is the synthesis of the two subunits in comparable amounts and their encounter at the appropriate time; that is, the formation of heterodimers should be competitive with the formation of “unwanted” homodimers. When aiming to turn a homodimeric protein into a heterodimeric complex, it is important to know which factors influence this process to tweak the dimerization in favor of heterodimer formation. Moreover, a deeper understanding of the mechanisms of (membrane) protein multimerization is important from a general scientific point of view. 

Spatial separation of the sites of protein synthesis may be a key factor in the multimerization of proteins. For luciferase in *E. coli*, it has been shown that the efficiency of dimerization of its subunits (LuxA and LuxB) decreases when transcribed from distant chromosomal sites [12]. In a different study [38], Shiber and colleagues found that nine out of 12 hetero-oligomeric protein complexes in *Saccharomyces cerevisiae* assemble co-translationally, and the other three make use of chaperones, illustrating that assembly of oligomers is a complex process. 

Membrane proteins have a lower lateral mobility than cytoplasmic proteins [39,40] and are restricted to a two-dimensional space. It is very well possible that the dimerization of membrane proteins has an even stronger dependence on the location of synthesis. The self-association of proteins in membranes is enhanced by their orientational restriction and volume exclusion as a result of the macromolecular crowding in the membrane [14]. When the genes for heteromeric complexes are not present in a single operon, it is likely that the local concentration of the identical protomers (produced from one transcript) is higher than that of dissimilar protomers (produced from different transcripts). Thus, the chance of two proteins to assemble into a complex will be higher when synthesized from a single transcript and polysomes docked onto vicinal Sec translocons than when the proteins are produced from distant transcripts. With this reasoning, we would expect to find less heterodimeric OpuA, because OpuABC-H and OpuABC-SS are synthesized from different transcripts. Indeed, we see that most of the protein is lost during the two-step purification, even after optimizing the process. This can indicate that there is room for improvement in the purification protocol, but we believe that the data hint at the presence of large amounts of homodimeric protein. Yet, we do find amounts of heterodimeric OpuA that are sufficient for follow-up studies. It is possible that the high expression levels of OpuA increased the chance for OpuABC-H and OpuABC-SS to find each other even when they are introduced into the membrane at distant sites.

The assembly of multimeric proteins is highly dependent on the proximity of their subunits. Once the folding of the protein chains begins, assembly can already occur between interface residues, increasing the risk of mis-assembly if the folding was still partial or defective. Therefore, many homomeric proteins are enriched for protein contacts toward their C-terminus, decreasing the tendency of premature assembly [41]. We reason that provided a structure is available, the formation of heterodimers can be enhanced by modification of the interaction region to create a complementary contact interface that favours heterodimer formation and prevents homodimerization. Alternatively, expressing the protomers in close proximity (perhaps on the same mRNA) to increase their local concentration could enhance the heterodimer formation.

While other factors can influence the formation of heterodimers theoretically, we have shown that a simple systematic approach involving the selection of the appropriate affinity tag and varying the temperature and production conditions can already significantly increase the heterodimer yield. As the overproduction of membrane proteins induces stress (vide supra), we reasoned that lowering the temperature may reduce the amount of protein and the amount of stress. We find the best conditions to be growth at 30 °C and a slow induction at 21 °C for a period of 4 h. 

Even though there is a wealth of information left to uncover on the formation of (hetero)multimeric proteins, we have paved the path to in vivo heterodimer formation out of a homodimeric protein and present an efficient way of purifying the formed heterodimers. The approach is feasible with relatively unstable proteins as the complexes are stabilized in the lipid environment of nanodiscs, i.e., after the first purification step. Further research may lead to improvements in the ratio of hetero- over homodimers and open the gate to the elucidation of complex formation in vivo.

## 4. Materials and Methods

### 4.1. Materials

Common chemicals were ordered from Merck (Darmstadt, Germany) The pMSP1D1 plasmid was purchased from Addgene (Watertown, MA, USA) [20061]. The lipids 1,2-dioleoyl-sn-glycero-3- phosphatidylcholine (DOPC) [850375C], 1,2-dioleoyl-sn-glycero-3-phos- phatidylethanolamine (DOPE) [850725C], and 1,2-dioleoyl-sn-glycero-3-phosphatidylglycerol (DOPG) [840475C] were purchased from Avanti Polar Lipids, Inc (Alabaster, AL, USA) (>99% pure, in chloroform). n-dodecyl-β-Dmaltoside (DDM) [D97002] was purchased from Glycon Biochemicals GmbH (Luckenwalde, Germany).

### 4.2. Construction of Strains and Growth Conditions

The bacterial strains, plasmids, and primers used in the present study are listed in Table 1. Plasmids were propagated in *Lactococcus lactis* strain Opu401 [26] (which is *L. lactis* NZ9000 with the *opuA* genes deleted). To construct the OpuA StrepII-tagged homodimer, the *pNisA* promoter [41] and the *opuABC* and *opuAA* genes were PCR-amplified from pNZopuAHis [26] with primers 6428 and 6429. The backbone of the pIL253 vector [24] was amplified with primers 6430 and 6431, which contained the StrepII-tag sequence. The two amplified fragments were ligated to create the pILopuAS vector. The OpuA TwinStrepII-tagged homodimer was constructed using the pILopuAS vector as a template, where the StrepII-tag polypeptide sequence (SA-WSHPQFEK) was exchanged for the TwinStrepII-tag (WSHPQFEKGGGSGGGSGGS-SAWSHPQFEK), yielding the pILopuASS vector. As a result, the three vectors contained the *opuAA* and *opuABC* genes under the control of the nisin-inducible *pNisA* promoter (the original operon structure was retained, except that the native *opuA* promoter was replaced by *pNisA*), and the recombinant genes produce proteins with an affinity tag at the C-terminus of the OpuABC subunit. The strains producing the heterodimeric OpuA carry two possible combinations of plasmids: pNZopuAHis with pILopuAS or pNZopuAHis with pILopuASS. To construct the *recA* deletion strain *L. lactis* 401*ΔrecA*, the flanking regions of *rec**A* were amplified using primers 7036 and 7037. The pCS1966 vector [42] was amplified with primers 7038 and 7039. These two fragments were ligated, obtaining the pCS1966-RecA vector. The pCS1966 derivative was obtained and maintained in *Escherichia coli* K-12 strain MG1655 [43]. The pCS1966-RecA vector was introduced in *L. lactis* 401 cells, and positive colonies were selected in SA medium plates [44] supplemented with 20 μg/mL 5-fluoroorotic acid hydrate. The double knockout strain (*∆opuA, ∆recA*) was named *L. lactis* 401*ΔrecA*.

All the constructs were engineered by the ligation-free uracil-excision based-technique USER cloning method [46]. The PCR amplifications were carried out by the PfuX7 DNA polymerase [47] with uracil-containing primers, and the amplified fragments were ligated with USER enzyme (New England Biolabs, Inc, Ipswich, MA, USA). Plasmid DNA was isolated using the Macherey-Nagel NucleoSpin Plasmid QuickPure Kit (Thermo Fisher Scientific, Groningen, The Netherlands). DNA clean-up was performed with the Macherey-Nagel NucleoSpin Gel and PCR Clean-up Kit (Thermo Fisher Scientific, Groningen, The Netherlands). Constructs were checked by PCR amplification and subsequent sequencing analysis by Eurofins Scientific (Heerenveen, The Netherlands). The strains were routinely cultivated semi-anaerobically at 30 °C in M17 broth (Oxoid, Wesel, Germany) supplemented with 1% (*w/v*) glucose (Merck, Darmstadt, Germany). When needed, the media were supplemented with 5 μg/mL of erythromycin and/or chloramphenicol.

### 4.3. Expression of OpuABC Genes

*L. lactis* OpuA-producing strains were grown in 2 or 10 L pH-controlled bioreactors. Cells were propagated in M17 broth supplemented with 1% (*w/v*) glucose plus 5 µg/mL of the appropriate antibiotic at 30 °C with stirring (200 rpm). A constant pH of 6.5 was kept by titrating the culture with 4 M KOH. Unless specified otherwise, cultures were induced at an OD_600_ of 2 with 0.05% (*v/v*) of nisin A* (culture supernatant of the nisin A producing strain *L. lactis* NZ9700 [36]) to initiate the transcription from the *NisA* promoter and the addition of extra 1% (*w/v*) glucose to obtain higher growth yields. To promote heterodimer formation, the temperature of the culture was decreased to 21 °C, during the induction, by cooling the bioreactor with ice-cold water, simultaneously with addition of the inducer. The decrease of temperature from 30 to 21 °C took approximately 15 min. Cells were harvested by centrifugation (15 min, 6000× *g*, 4 °C) after 2 h of induction, washed twice, and resuspended to an OD_600_ of 100 in ice-cold 50 mM KPi pH 7.5 buffer, flash-frozen, and stored at −80 °C. 

### 4.4. Optimization of the Induction Conditions

To promote the heterodimer formation in the cell, different induction conditions were tested; that is, nisin A* concentration, the post-induction temperature, and induction time were varied. For this purpose, *L. lactis* Opu401, carrying pNZopuAHis and pILopuAS, was grown in a 2 L flask containing M17 broth supplemented with 1% (*w/v*) glucose, 5 μg/mL erythromycin, plus 5 μg/mL chloramphenicol at 30 °C with stirring (200 rpm). When the culture reached an OD_600_ of 0.5, it was divided into smaller cultures of 50 mL each and induced with 0.05%, 0.02%, 0.01% or 0.002% (*v/v*) of nisin A*. Then, the cultures were incubated at 21 °C or 30 °C for induction times varying from 2 to 8 h. Cells were harvested by centrifugation (15 min, 6,000× *g*, 4 °C), washed twice, and resuspended to an OD_600_ of 34 in ice-cold 50 mM KPi, pH 7.5. Samples of 1.5 mL were mixed with 400 mg of 0.1 mm glass beads (Merck, Darmstadt, Germany) and lysed with a TissueLyser LT (Qiagen, Hilden, Germany) for 5 min at high speed. Glass beads and cellular debris were removed by centrifugation (15 min, 25,000 x g, 4 °C). Pellets were discarded, and membrane vesicles were collected by centrifugation (20 min, 267,000 × *g*, 4 °C) and resuspended in 1.8 mL of ice-cold 50 mM Kpi pH 7.5 supplemented with 20 % (*v/v*) glycerol. Then, membrane vesicles were solubilized with 0.5% (*w/v*) DDM and nutated for 2 h at 4 °C, after which Ni^2+^-Sepharose purification was carried out as described below. The presence of the heterodimer was analyzed by immunoblotting the elution samples with antibodies against StrepII-tag.

### 4.5. Isolation and Preparation of Membrane Vesicles

The isolation and preparation of membrane vesicles were performed as described in [48] with minor changes. Cell pellets were thawed on ice and supplemented with 2 mM MgSO_4_ plus 100 μg/mL DNAse. Cells were lysed by double passage through a cell disruptor (Constant Systems Ltd., Daventry, UK) at 39,000 Psi. After lysis, 1 mM PMSF plus 0.05 M EDTA (pH 8.0) were immediately added to avoid protein degradation. The cell debris was removed by centrifugation (15 min, 12,000× *g*, 4 °C), and the membrane vesicles were collected by ultracentrifugation (1 h, 267,000× *g*, 4 °C) and resuspended in ice-cold buffer A (50 mM KPi, pH 7.5, 20 % (*v/v*) glycerol). Aliquots were flash frozen and stored at −80 °C. The protein concentration was determined using the Pierce BCA Protein Assay Kit (Thermo Fisher Scientific, Groningen, The Netherlands).

### 4.6. Purification of OpuA

The purification process of OpuA was divided in a series of steps, of which the order was determined by the type and number of affinity tags of the final construct. Hence, the homodimeric forms of OpuA were subjected to a single affinity purification depending on the affinity tag, that are Nickel-Sepharose or Strep-tactin purification, followed by reconstitution in nanodiscs and size exclusion chromatography on a Superdex 200 increase 10/300 GL column. However, the heterodimeric forms of OpuA required a two-step purification process that could be conducted in different order as described in the Results section. Below, we independently describe all the required steps for the purification of any of the OpuA constructs but note that the order may vary for each of them. For the solubilization of membrane vesicles, prior to Ni^2+^-Sepharose affinity purification, membrane vesicles were quickly thawed and diluted to a final protein concentration of 5 mg/mL in 50 mM KPi pH 7.0, 200 mM KCl, 20% (*v/v*) glycerol plus 10 mM imidazole. When Strep-tagged protein samples were to be purified, membrane vesicles were harvested (20 min, 267,000× *g*, 4 °C) and pellets were dissolved in 50 mM Tris-HCl pH 8.0, 150 mM NaCl plus 20% (*v/v*) glycerol at a protein concentration of 5 mg/mL. Then, membrane vesicles were solubilized with 0.5% (*w/v*) DDM and nutated for 1 h at 4 °C. Supernatant was collected by ultracentrifugation (20 min, 267,000× *g*, 4 °C).

### 4.7. Ni^2+^-Sepharose Affinity Purification of His-Tagged Proteins

Ni^2+^-Sepharose resin (GE Healthcare, Hoevelaken, The Netherlands) (0.5 mL of resin per 10 mg total protein) was pre-equilibrated with 12 column volumes (CV) of distilled water followed by 4 CV of wash buffer (50 mM KPi pH 7.0, 200 mM KCl, 0.02% (*w/v*) DDM plus 20% (*v/v*) glycerol) supplemented with 10 mM imidazole (pH 7.5). To decrease the detergent concentration, solubilized membrane vesicles were diluted five-fold in ice-cold buffer (50 mM KPi pH 7.0, 200 mM KCl, 20% (*v/v*) glycerol plus 10 mM imidazole) and then incubated with the Ni^2+^-Sepharose resin under rotation for 2 h at 4 °C. The mixture was poured into a column, and the resin was washed with 20 CV of wash buffer supplemented with 50 mM imidazole. Proteins were eluted with 2.5 CV of wash buffer supplemented with 500 mM imidazole. Protein concentration in the elution fractions was determined by absorbance measurements at 280 nm.

### 4.8. Reconstitution in Lipid Bilayer Nanodiscs

Reconstitution was performed as described previously [27] with some modifications. Synthetic lipids were mixed in a ratio of 50 mol % DOPE, 12 mol % DOPC and 38 mol % DOPG, and the preformed liposomes were prepared as described by [49]. Then, the mixture was extruded 13 times through a 400 nm polycarbonate filter (Avestin Europe GmbH, Mannheim, Germany) to obtain large unilamellar vesicles and then solubilized with 12 mM DDM followed by heavy vortexing. The standard procedure for the reconstitution of OpuA in nanodiscs was at an OpuA/MSP1D1/liposomes ratio of 1:20:2000 (*w/w*), respectively, in a final volume of 700 μL giving the following composition: 50 mM KPi pH 7.0, 12 mM DDM, 4% (*v/v*) glycerol, 0.72 μM OpuA, 14.3 μM MSP1D1 plus 1.43 mM lipid. To optimize the method, we tested an OpuA/MSP1D1/lipid ratio of 1:20:1000 (*w/w*), starting with a 6 times higher concentration of OpuA, having a final composition of 50 mM KPi pH 7.0, 12 mM DDM, 4% (*v/v*) glycerol, 4.32 μM OpuA, 85.8 μM MSP1D1 plus 4.29 mM lipid. When needed, OpuA samples were concentrated in 0.5 mL 30,000 kDa concentrators (Vivaspin). The reconstitution mixture was nutated for 1 h at 4 °C, after which detergent was removed by adding 500 mg of SM2 Biobeads and incubating overnight at 4 °C with gentle agitation. Biobeads and protein aggregates were carefully removed by transferring the sample with a syringe to a new Eppendorf tube and subsequent centrifugation (25,000× *g*, 10 min, 4 °C). Then, the mixture was fractionated by size exclusion chromatography (SEC), using a Superdex 200 Increase 10/300 GL column (GE Healthcare, Hoevelaken, The Netherlands) equilibrated with 50mM KPi pH 7.0 supplemented with 200 mM KCl. Protein-containing fractions were pooled and stored at 4 °C until further use.

### 4.9. Purification of Strep-Sagged OpuA Proteins

Strep-tactin superflow high-capacity resin (IBA LifeSciences, Göttingen, Alemania) (1 mL resin per 10 mg/mL protein) was pre-equilibrated with 4 CV of buffer W1 (50 mM Tris-HCl pH 8.0, 150 mM NaCl supplemented with 20% (*v/v*) glycerol plus 0.02% (*w/v*) DDM, for membrane vesicles samples) or with buffer W2 (50 mM Tris-HCl pH 8.0, 150 mM NaCl supplemented with 4% (*v/v*) glycerol, for samples containing OpuA nanodiscs). Samples were incubated with the pre-equilibrated resin and nutated for 1 h at 4 °C. The flow through was slowly passed twice through the column by gravity, and then, the column was washed 5 times with 3 CV of buffer W1 or W2. Proteins were eluted with 0.5 CV of buffer W1 or W2, both supplemented with 10 mM of d-Desthiobiotin. After 5 min of incubation, the elution was collected, and the procedure was repeated 4 times more. Protein concentration was determined in the elution fractions by absorbance measurements at 280 nm.

### 4.10. SDS-PAGE and Western Blotting Analysis

Samples from all the steps of the purification process were collected and analyzed by SDS-PAGE using 12.5% poly-acrylamide gels. Pictures of the Coomassie-stained gels were taken by a Fujifilm LAS 3000 Imaging system (Fujifilm, Düsseldorf, Germany). To confirm correct subunit composition, Western blot analyses were carried out. Samples were resolved in a 12.5% SDS-PAGE and transferred to a PVDF membrane with primary antibodies against StrepII-tag or His_6_-tag (Qiagen, Hilden, Germany). Transfer of the proteins was done in 40 min at 0.08 A in a Trans-Blot SD Semi-Dry Transfer system (Bio-Rad, Nazareth, Belgium). Proteins were visualized by inducing chemiluminescence with the CDP-star kit (tropix, inc., London, UK) in the LAS-3000 imaging system.

### 4.11. ATPase Activity Assay

The ATPase activity of OpuA reconstituted in nanodiscs was analyzed using a coupled enzyme assay as described previously [27,50]. In brief, the measurements were performed at 30 °C in a 96-well plate using a Spark 10 M 96-well plate reader (Tecan, Männedorf, Suiza). A standard measurement solution of 200 μL per well contained 50 mM KPi (pH 7.0), 0.3 M KCl, 57 nM OpuA reconstituted in nanodiscs, 4 mM sodium phosphoenolpyruvate, 0.3 mM NADH, and 3.5 μL of pyruvate kinase/lactic dehydrogenase enzyme mixture from rabbit muscle in 50% glycerol, with or without 62 μM substrate (glycine betaine). After incubation for 3 min at 30 °C, 10 mM MgATP pH 7.0 was added to each well, and the absorbance of NADH at 340 nm was monitored over a period of 15 min. The oxidation of NADH is stoichiometrically coupled to the amount of ATP consumed, and the ATPase activity was expressed as the moles of ATP hydrolyzed per min per mg of OpuA.

## Figures and Tables

**Figure 1 ijms-22-05912-f001:**
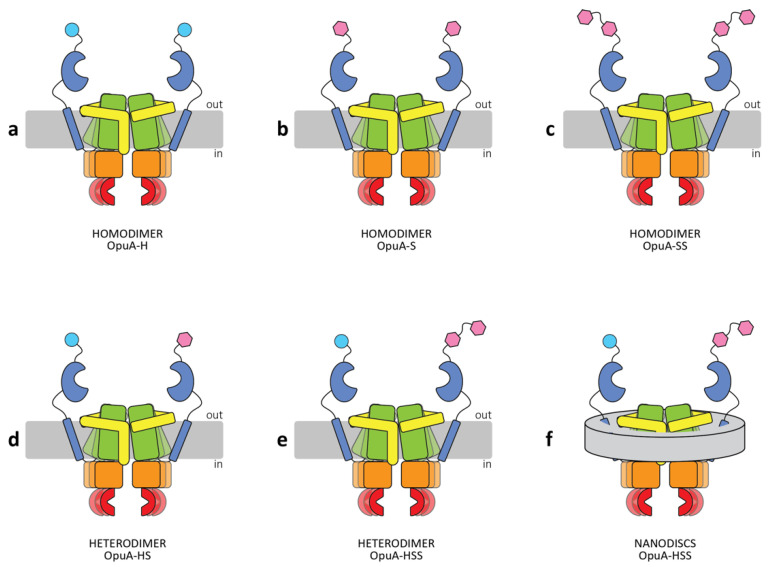
Schematic representation of various OpuA constructs used in this study. The wild-type transporter OpuA is composed of two OpuAA subunits, each carrying a tandem cystathionine-β-synthase (CBS) domain (red) and the ATP-binding domain (orange), and two OpuABC subunits, each carrying a transmembrane domain (TMD) (green), including the scaffold domain (yellow) and the substrate-binding domain (SBD) (blue). (**a**) Homodimeric OpuA-H, the wild-type OpuA, with a His_6_-tag (cyan circle) linked to the SBD; (**b**) Homodimeric OpuA-S, OpuA tagged with a StrepII-tag (pink hexagon) linked to the SBD; (**c**) Homodimeric OpuA-SS, OpuA containing a TwinStrepII-affinity tag (double pink hexagon) linked to the SBD; (**d**) Heterodimeric OpuA-HS, OpuA containing a His_6_-tag in one SBD and a StrepII-tag in the other SBD; (**e**) Heterodimeric OpuA-HSS, OpuA composed of one SBD tagged with His_6_-tag and another one with TwinStrepII-tag; (**f**) Schematic representation of OpuA-HSS in nanodiscs; lipids and MSP1D1 scaffolding protein are shown as grey discs.

**Figure 2 ijms-22-05912-f002:**
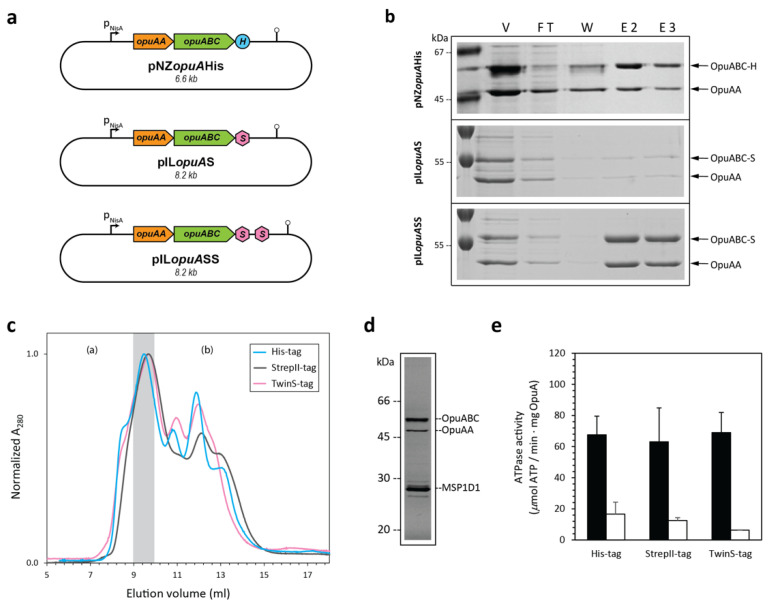
Characterization of three differently tagged homodimeric OpuA constructs. (**a**) Schematic plasmid maps of the expression vectors. *OpuAA*, gene encoding the ATPase subunit and CBS domains of OpuA; *opuABC*, gene encoding the TMD and SBD of OpuA; *pNisA*, nisin-inducible promoter; H, His_6_-tag; S, StrepII-tag; double S, TwinStreptII-tag; bent arrows and lollipop symbols represent the promoters and terminators, respectively. (**b**) SDS-PAGE analysis (12.5% polyacrylamide) of affinity purifications of the three homodimeric OpuA constructs (OpuA-H, OpuA-S, and OpuA-SS). The indicated proteins were purified from crude membrane extracts as explained in the text. The fractions tested were membrane vesicles (V), column flow through (FT), wash (W), and elution fractions (E). (**c**) Size exclusion chromatography profiles of homodimeric OpuA nanodiscs, using a Superdex 200 increase 10/300 GL column. The chromatograms were normalized to the highest peak. The peak fractions used for further analysis are indicated by the gray shading. (a) and (b) represent the peak fractions of aggregated and empty nanodiscs, respectively. (**d**) Typical peak fraction of nanodiscs analyzed by 12.5% SDS-PAGE, showing the presence of OpuAA, OpuABC, and the scaffold protein MSP1D1. (**e**) ATPase activity in the presence (black bars) and absence (white bars) of 62 μM substrate (glycine betaine). Error bars represent the standard deviation of independent triplicates.

**Figure 3 ijms-22-05912-f003:**
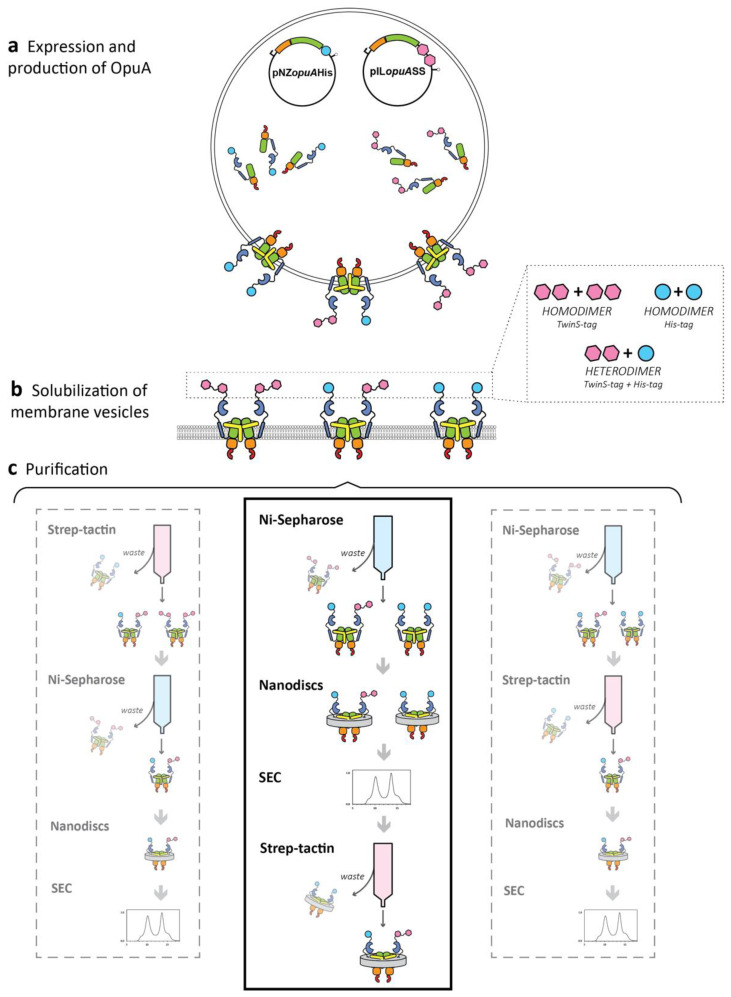
Schematic of the purification of the heterodimeric OpuA. (**a**) *L. lactis* Opu401 strain carrying plasmids pNZopuAHis and pILopuASS was grown in glucose-M17 broth at 30 °C and the genes were expressed with 0.05% (*v/v*) nisin A* at 21 °C for 4 h. Three possible OpuA variants are formed in the cell: OpuA-H, OpuA-SS, and OpuA-HSS. (**b**) Solubilization of membrane vesicles were carried out as described in the Methods section. (**c**) Three different purification strategies were tested by varying the order of the different steps. The central bold lined square highlights the most efficient protocol.

**Figure 4 ijms-22-05912-f004:**
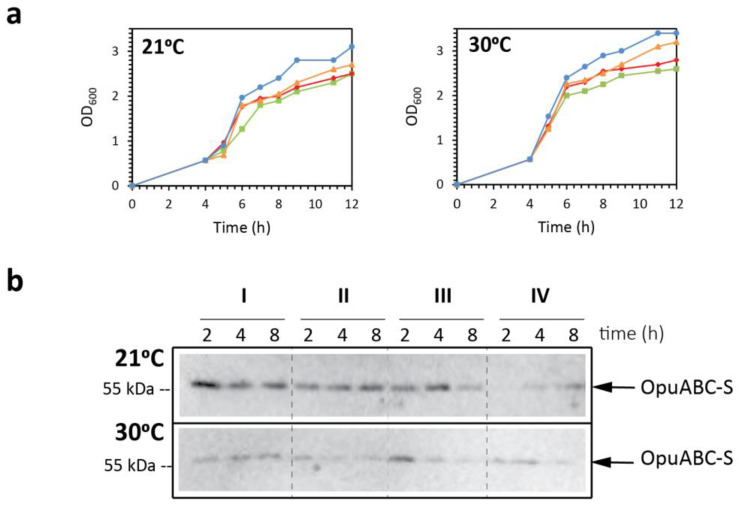
Optimization of the heterodimer formation under different induction conditions. (**a**) *L. lactis* Opu401 strain, harboring plasmids pILopuAS and pNZopuAHis, was propagated at 30 °C in glucose-M17 broth as described in the text. When cultures reached an OD_600_ of 0.5, they were induced at four different nisin A* concentrations: 0.05% (green square), 0.02% (red diamond), 0.01% (orange triangle), and 0.002% (blue circle). Then, cultures (50 mL) were incubated at two different temperatures, 21 or 30 °C, and induction times of 2, 4, and 8 h were tested. (**b**) Membrane vesicles were obtained, and proteins were purified with Ni^2+^-Sepharose resin. To check the presence of the heterodimeric OpuA variant, final elution fractions were analyzed by Western blot analysis, using monoclonal antibodies directed against the StrepII-tag. The Roman numerals indicate the different nisin concentrations: 0.05% (I), 0.02% (II), 0.01% (III), and 0.002% (IV).

**Figure 5 ijms-22-05912-f005:**
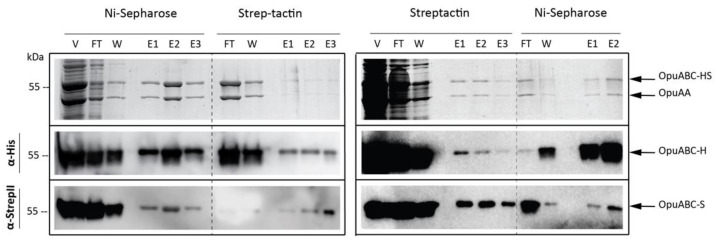
SDS-PAGE (upper panel) and Western blot (two lower panels) analysis of the two-step affinity purification of OpuA. *L. lactis* Opu401 carrying plasmids pNZopuAHis plus pILopuAS was grown and induced under the following conditions (0.01 % nisin A*; 21 °C during induction; 4 h of induction). Membrane vesicles were obtained as described in the Methods section, and after solubilization of the membranes with 0.5 % (*w/v*) DDM, the lysate was subjected to two affinity purification steps: Ni^2+^-Sepharose followed by Strep-tactin (left panel) or vice versa (right panel). The following fractions were tested: vesicles (V), flow through (FT), wash (W), and elution fractions (E). Monoclonal antibodies directed against the His_6_-tag and StrepII-tag were used, as indicated on the left side of the immunoblots.

**Figure 6 ijms-22-05912-f006:**
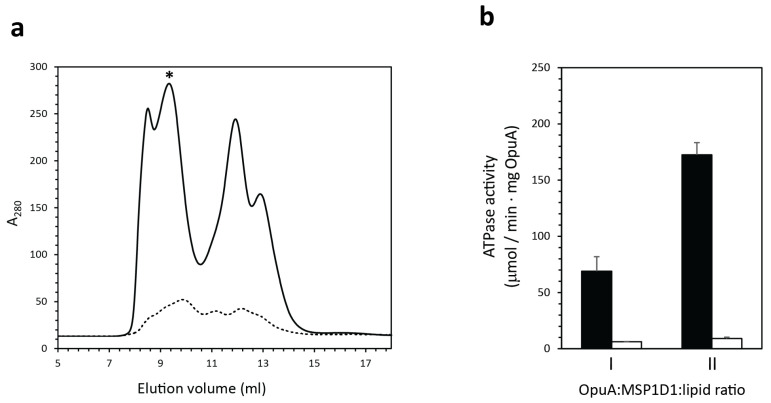
Optimization of OpuA reconstitution in nanodiscs. Homodimeric OpuA-SS was purified and reconstituted in nanodiscs formed at different molar ratios and concentrations. (**a**) Size exclusion chromatography profile of nanodiscs formed at a OpuA/MSP1D1/lipids ratio of 1:20:2000 (dotted line) and 1:20:1000 ratio (solid line); in the latter case, we used a six-times higher concentration of OpuA. Star represents the peak fraction with active OpuA-SS nanodiscs. We verified the dimeric state of OpuA in the peak fraction from the intensity of OpuAA, OpuABC, and MSP1D1 bands on SDS-PAA gels. (**b**) ATPase activity of OpuA-SS reconstituted in nanodiscs formed at a ratio of 1:20:2000 (I) and 1:20:1000 (II). Black and white bars represent activity in the presence and absence of 62 μM glycine-betaine, respectively. Error bars represent the standard deviation of triplicates.

**Figure 7 ijms-22-05912-f007:**
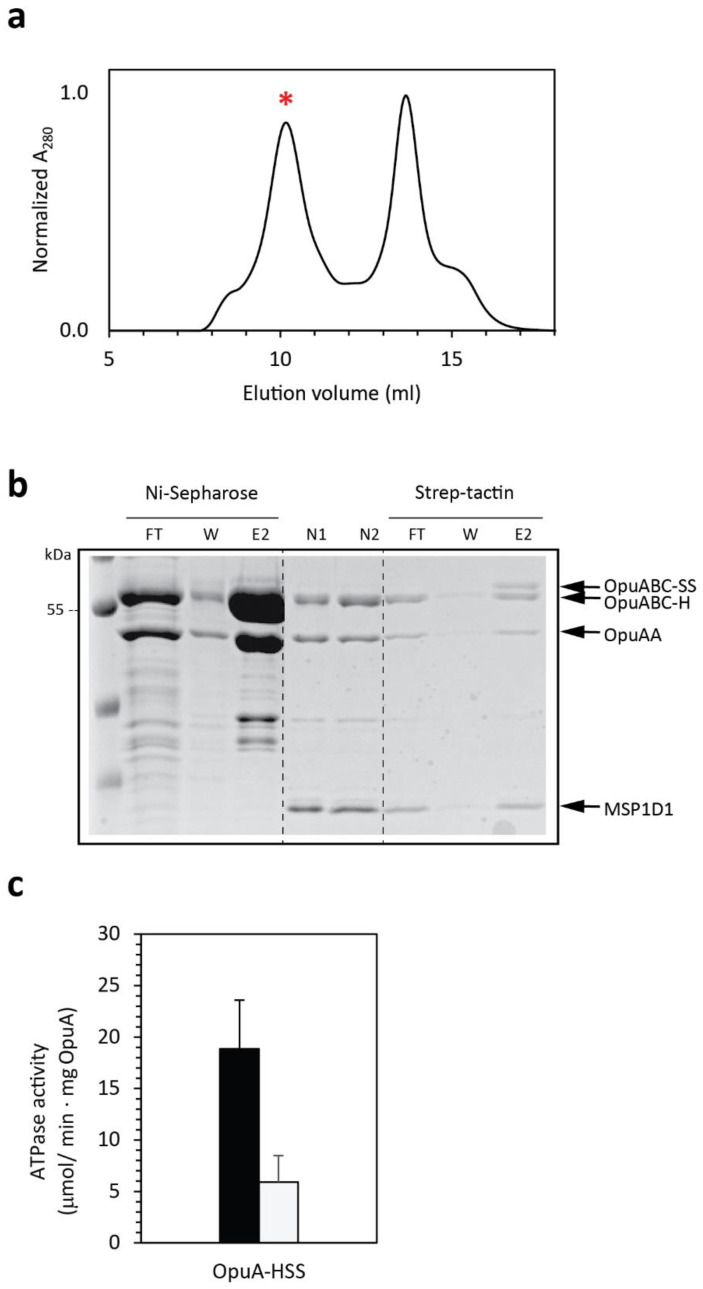
Purification and characterization of the heterodimeric OpuA-HSS. Membrane vesicles containing a mixture of OpuA-H, OpuA-SS, and OpuA-HSS were obtained as described in the Methods section and subjected to a series of purification steps: (i) Ni^2+^-sepharose purification; (ii) nanodisc reconstitution; (iii) size exclusion chromatography; and (iv) Strep-tactin purification. (**a**) Size exclusion chromatography profile of the OpuA-H and OpuA-HS nanodiscs. Star represents the peak fraction used for further studies. (**b**) Coomassie-stained 12.5% SDS-PAGE samples of the different stages of the purification process. The fractions tested were flow through (FT), wash (W), elution (E), and the peak fraction containing nanodiscs (N). Note that OpuABC-H and OpuABC-SS subunits can be distinguished by their different migration in 12.5% SDS-PAGE gels. (**c**) ATPase activity in the presence (black bar) and absence (white bar) of 62 μM glycine-betaine. Error bars represent the standard deviation of independent triplicates.

**Table 1 ijms-22-05912-t001:** Strains, plasmids, and oligonucleotides used in the present study.

Strain	Reference
*L. lactis* Opu401	[26]
*L. lactis* NZ9700	[45]
*L. lactis* 401*ΔrecA*	This work
*E. coli* MG1655	[43]
**Plasmid**	**Reference**
pNZ*opuAHis*	[26]
pIL*OpuAS*	This work
pIL*OpuASS*	This work
pCS1966	[42]
pCS1966-RecA	This work
pIL253	[24]
pMSP1D1	Addgene
**Oligonucleotide**	**Sequence**
6494	AATCGATAAGCTTGGCTGCAG
6493	AACGAAGTGAGGGAAAGGCTAC
6429	AAACTGCGGAUGAGACCAAGCAGAACGACCCTCAATGGATCC
6428	AGCTCCAAGAUCTAGTCTTATAAC
6430	ATCCGCAGTTUGAAAAATAATAATTGGATTAGTTCTTGTGGTTACG
6431	ATCTTGGAGCUTCCATGTAATCGGGTTCTTC
7288	AATCAATCAUGAACCTGCTCCTC
7287	ATGATTGATUGGATTAGTTCTTGTGGTTACG
7234	AATTCCCAAGUTAGTCATTCTGACTG
7233	ACTTGGGAATUCGTCAAGTTTCAACGGAATTAG
7038	AGGCTACACTAGUTCTAGAGCG
7039	AGGTTGTCCACUCGGTACCCAG
7037	ACTAGTGTAGCCUTCAAGATCCTAGTCAGCATTCC
7036	AGTGGACAACCUATAGAAGCCACTTATCCAAG

## Data Availability

All data needed to evaluate the conclusions in the paper are present in the paper. All data and materials used in the analysis are available upon request to the lead author.

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
