# Peer review of "Heterodimer Formation of the Homodimeric ABC Transporter OpuA"

_ijms, 2021, doi:10.3390/ijms22115912_

Round 1
Reviewer 1 Report
The manuscript by Alvarez-Sieiro et al. presents an in vivo approach for the isolation of homodimeric proteins as ‘hetero’dimeric proteins composed of modified homodimers by the use of affinity tagged variants. ‘Hetero’dimeric proteins allow studies on the interaction and relation of monomers in homooligomeric proteins and multimerization of membrane proteins. The method is tested for the well characterized OpuA ABC transporter. A carefully selected approach of established methods is used to produce the protein with significant yields as a heterooligomer. The study is carefully designed and executed based on established approaches. The study demonstrates that expression conditions are crucial, as well as the order of the purification steps and the reconstitution in nanodiscs. The approach was useful for the isolation of heterodimers of OpuA, the general applicability has to be tested.
The approach relies on the duplication of the opuAA opuABC operon (with different tags) on separate plasmids. The authors suggest on line 337 the chance of two proteins to assemble into a complex will be higher when synthesized from a single transcript. This suggestion raises the question why the opuABC gene was not duplicated in one operon and one plasmid (eg opuAA-opuABC-opuABC, with suitable tags) to obtain optimal yield of the heteromultimeric complex (OpuAA)2-(OpuABC-H)-(OpuABC-S).
The authors discuss in the experiments of Fig. 4 the fraction of the Opu-HS heteroform. For a better estimation of the fraction, OpuABC-H contents and the total protein should be tested as well by Westernbloting with anti-His and by an SDS-PAGE.
The recovery of the OpuA-HSS heterodimer appears to be very low (0.2%). How does this compare to the recovery for other heterodimers under comparable conditions?
Author Response
The comments are addressed in the attached rebuttal.

Reviewer 2 Report
General Comments: This paper presents a method for affinity purifying the same subunit of the ABC transporter OpuA with different affinity tags for the purpose of differentiating the otherwise identical homomeric subunits for future SAR experiments and without the requirement for attaching probes to the subunits for smFRET, DEER or equivalent techniques. Overall, it is very well presented and does utilise a relatively novel approach for homodimer differentiation (basically by turning them into heterodimers). However, the main issue with the paper is the lack of success in the stated aim as I'm sure the authors are aware. The functional recovery and yields of any heterodimer OpuA combination were poor. The inclusion of additional data not shown and results that could be shown will improve the rigour of the paper despite the difficulties encounted. I do commend the researchers on their attempts at optimisation and think the paper is well worth publishing in IJMS with some revisions. It is not easy to overcome expression difficulties with membrane proteins, especially multimeric ones.
A few general things to correct and watch out for:
- There are numeroud spelling mistakes and inconsistencies in the notation and spelling: check the document carefult before re-submission and provide and abbreviation list for 4-letter lipid names etc.
- I became a little lost at times with the lack of figures for certain results and the dis-jointed nature of the presented results (see specific comments below). Please be more specific in linking results to specific figure panels and explicitly stating results clearly. Also, do not allow results and figures to jump out of order - either state and present results and their accompanying figure in order or do not mention them until required. I refer mostly to figure 5-7 below.
Introduction
- A very well written and comprehensive introduction overall; I wish there were more like this. Perhaps the only comment I have is to point out, while I wouldn't disagree with any of the points made, there are additional, more recent and comprehesive reviews on the subject of homo- and hetero-multermeric protein formation and biology that focus on membrane proteins that would be more appropriate for you to reference. The knowledge of the authors is good but these reviews (see below) may further inform the introduction and subsequent discussion of the findings in the wider context of multimeric membrane protein formation. In particular, these reviews may be more relevant than some of your current references as they are specific for membrane proteins and bring the trends in the field up-to-date, including the specific intracies of different membrane protein homo- and hetero-multimer formation and function. Good addition, recent papers include: DOI: 10.1007/5584_2020_584 DOI: 10.1080/09687680802712422 DOI: 10.1146/annurev-biochem-060409-092524 DOI: 10.1146/annurev-biochem-013118-111947
- The same point holds for the introduction section were you cite references 11 and 12 - the above reviews are better for the reasons stated.
- For the sentence 'The OpuA complex in the detergent-solubilized state disassembles into two OpuABC and two OpuAA subunits when the glycerol concentration falls below 15-20% (v/v), but the dissociation of the complex is reversible.' Please specify which detergent; is it a generic membrane preparation solubilised with a high CMC detergent like DDM as is common or something more specific. Furthermore, what would be the reason such a high glycerol concentration stabilises dimer formation? This is above what is typically used for elution buffers during typical mem protein purifications steps, which are usually less than 10% glycerol. This may be of interest to the protein chemist readers.
Results
- In the first sentence I guess you mean expression plasmids for L.lactis ,right? It might be worht explicitly stating this to avoid confusion.
- What is the number of His residues in the tag? Typically we might write His6 or His9 etc when we first mention the tag. I would add the subscript to denotate the lenght of the His tag wherever it is written.
- Figure 2b: is a little unusual in that (without knowing your previous work and purification of OpuA) you observe both OPuABC and OpuAA His-tagged subunits in all fractions of the SDS-PAGE, including the flowthrough and the wash. To me, this suggests binding to the Ni-sepharose resin is not of very high affinity. Have you observed this before with OpuA tagged with Histidine tags? Was this, therefore, expected? It also appears as though there is a subunit effect, where OpuAA is eluted in all fractions to the similar extent, whereas OpuABC does predominate as might be expected, in the elution fractions. If His-tag purification is most efficent, as you state in the text, is still seems like you're loosing a lot in the non-elutant fractions and the purification yield could still be improved - this may also be worth a discussion point. Please explain in text and provide and relevant references to address these questions and to clarify these points.
- Figure 2b: Still on this figure. Is there any reason for not showing all the elution factions from the imadazole elution? I can see that you only have the E2 and E3. Was there small amounts of the proteins coming off in other elution fractions (as I might suspect given the previous point)?
- Minor point: the 4-letter lipid abbreviations for lipids are not, in my experience, widely understood by non-lipid or non-membrane biochemist. Either provide an abbreviation list of give the pull chemical name when they arefirst used in the text.
- The sentence: 'The initial protocol to purify the hetero OpuA-HS mutant (Fig. 1) was based on a Ni2+-Sepharose purification to remove Strep-tagged homodimers (OpuA-S) followed by a Streptactin purification to remove His-tagged homodimers (OpuA-H) or vice versa.' Surely this is a typo and it is the other way around, yes?
- Okay, so you could not purify any combination of the 3 differentially tagged monomers in the background or the deltaRecA L.lactis strains (sections 2.2, 2.3). Fair enough, I have no problem with negative results and hypothesis testing. However, I would like to see some data on this. Please show an example of the negative SDS-PAGE and/or chromatographs for these two sections. You should be able to say in a little more detail why the protein heteromers were not forming by seeing (as you have conducted with Fig. 2) if the the protein was actually produced at all i.e. in the crude membrane vesicles. You only state, that I can see, that no protein was detected at the end of the 2-step purification (affinity+gel filtration) but not through the process. This would have also told you if you were getting any protein at all, the answer to which is implied by the optimisation in the next section. Alternatively, you could remove sections 2.2 and 2.3 as spearate sections and simply state that initial results fails and hence you moved to optimisation (section 2.4) - it seems strange that you labour with so much description (and a very nice schematic for fig 3) without showing any results. Either condense the results or show them.
- The rationale at the start of section 2.4 is okay - this is membrane protein yields 101 - but you have no hypothesis as to why specifically the combination of differentially tagged monomers in more than one expression vector would effect the overall expression? This seems to me to be the key point - you show single affinity tag transformation and expression in L.lactis is good (fig.2) but not when any combination of two expression vectores expressing two different tagged subnits is expressed. Why specifically would this change cause such a dramatic loss of yield?
- Figure 4b: I presume the 'I, II, III' etc notation above the SDS-PAGE signify the different nisinA concentrations as they do for Figure 4A, correct? Please make this explcit in the figure legend.
- Where you reference figures 3 and 4 in section 2.4 please remove the reference to figure 3 - only figure 4 shows the results referred to. Also, plese be more specific: refer specifically to 4A and 4B where relevant. At the moment the figure reference is too general and makes it a little difficult for the reader to locate the specific results you refer too.
- The sentence: 'The decrease in the temperature during induction reduced the final membrane vesicle protein yield from 55 to 37.5 mg/L but the amount of heterodimeric OpuA was increased by at least 2-fold.' Is this relative increase i.e. adjusted for the total protein yield, or an absolute increase? And where is the figure/calculations showing this? Or is it a summary statement of the coming results? If so it should be left until they have been stated.
- For Figure 5, the SDS-PAGE does support qualitatively what you state about the final purified heterodimers and total vesicle protein but thes are quantitative results. How ere they obtained? By denistometry from the SDS-PAGE or (far more accurately) by integrating the OD280 curves on the chromatography curves or with a total protein assay (e.g. BCAA for total vesicle protein)? Whatever the method, more details and/or figures to demonstrate the quantifications need to be provided.
- The text 'OpuA subunit (OpuABC-SS) not only allowed obtaining a higher protein yield, but this subunit also migrates differently on SDS-PAA gels and clear separation from the his-tagged subunit (OpuABC-H) (see example on Fig. 7),' Fair enough but either keep the results figures in order (in such case this becomes fig 6) or only mention the result where you want to present the data in a figure. This seems strange to mention a result in passing before moving onto another result.
- Please show the 'data not shown' figure for section 2.5. In fact this initial optimisation with the twin-strept tag needs a figure. As with the previous comment: it seems strange to present your first positive results in what was obviously a challenging prufication process only to show no figures. I would consider moving figure 7 to this sections and combining the additional text you have written later for that figure. To assist in this I would strongly suggest simply combining sections 2.5 and 2.6 into one - after all, these sections ahve the same aim and are showing the purification of the two-strept tagged OpuA.
- Figures 6 and 7 are actually some nice results following a tough optimisation process. One thing that escapes me is how you determine which of the SEC peaks is the fully formed homodimer in Figure 6 and Figure 7? There seems no attempt micelle calibration on the SEC to determine size of the nano-dics and no attempt to conduct ATPase assays on major eluted peaks, either of which might help clarify this point. Sorry if I missed something but this is not clear.
- Is glycine-betaine a substrate of OpuA ? I assume so, but please state it with reference when the ATPase assay is intriduced in the results.
- Please reference you figures in order - Figure 7B and 7A are presented out of order, either change the text or the figure so that Fig. 7A is presented and then Figure 7B.
- 'We attribute the somewhat lower activity of the heterodimer to the loss of a fraction of the OpuAA subunit as can be seen on SDS-PAGE gels (Figure 7A).' Lower activity compared to what? There is no other condition shown in this figure? Do you mean to the twin-strept purified homodimers in Figure 6.?
- I may be missing something again but I see from Figure 7A that it is elution fraction E2 which produces a band each for OpuABC monomers either His or twin-strept-tagged in the Strep-tatctin column. I see the same faint band in the E2 of the Ni-Speharose column elution. Yet I can not see OpuABC-SS in the Flow-through of the Strep-tatin column. This is confusing? By my reasoning you should be running the FT and/or wash fractions from the Ni-Sepharose purification through the Strep-tactin column, right? After all the twin-strept tagged- protein should not bind to the Ni column. Why then would you see a (very faint) band for OPUABC-SS in the E2 fraction of the Ni purification? It should not bind to the colum at all, yet I cannot see any stronger bands in any other Ni column fractions. The other possibility is that the heterodimer is intact throughout and I suspect this is what you are seeing. But then again, why do you not see any stronger band for the OpuABC-SS in the other (non-elution) Ni column fractions? None of this is clear in the results.
Discussion
Is good and makes the salient points. However, I would like you to focus a little more on why you have such a large apparent loss of yield in any case where you introduce heterdimeric expression of OpuA; at first with OpuABC-H and OpuABC-S and then with the combination of OpuABC-H and OPuABC-SS? The fact of the hetero-dimer itself and the large loss of purified protein seems to be the key bottle neck in the process. One point I would make that is not dicussed is the possibility that the simple expression of two separate expression vectors for mem proteins in L.Lactis itself might be casuing the loss of protein synthesis. From Figure 7 (se my last results comment) it seems a lack for OpuABC-SS is the issue in this attempted heterodimer expression. And in Figure 5 OpuABC-S seems to be by far the less expressed monomer. Are there examples of using two expression plasmids in L.lactis simultaneously? Are there other ways to express to membrane proteins (a heterodimer) in L.lactis simulataneously? i.e. multi-cistronic vectors? To improve the method and make it truely attractive as a way to purify and study homodimeric proteins what major variables in the biology of protein production need to rigorously optimised?
Author Response
Our response to the comments is given in the attached rebuttal.

Round 2
Reviewer 1 Report
No further recommendations
Author Response
Reviewer 1 had no further comments. We would like to thank her/him for the review.
Reviewer 2 Report
Thank you for addressing the reviewer comments. A few minor things still to attend to and some further clarification needed for one point:
- There are still some minor typos and grammatical errors. I found several with author names in the reference list. Whichever citation program you are using make sure it can import no standard characters.
- For point 10 of the results, if the dissociation of the OpuA has already been shown then please make this explicit in the text and cite the EMBO J paper referred to.
- Likewise, for results point 11, could the authors please add the references (Karasawa et al 2013; Sikkema et al 2021) to the text at the points at which this is mentioned and make the point that the nanodisc size has previously been determined explicit.
- Again, in point 11, I accept that the authors have run SDS-PAGE gels on the fraction corresponding to the elution volume at 9-10ml on the SEC and that them have determined the composition of this SEC fraction in another publication. I thank them for pointing this out in the figure 6 legend. However, to further clarify this point: how do the authors know that there is not significant amount of OpuA-SS-MSP1D1 nanodiscs coming out a the higher MW peak in the shown chromatograph or that smaller peaks do not contain dissociated OpuA subunits - thereby indicating significant dissociation of the full OpuA-SS during the nanodisc incorporation? Another way to ask the same question is what is the composition of the other 3 peaks in the chromatograph? Is the larger peak at ~8ml elution volume the void peak? The nature and composition of the 4 peaks observed in the SEC in Fig. 6 are important as the protein has been twin-affinity purified to this point and, therefore, is presumably quite pure. A clarification of the composition of each peak will allow the reader to have some estimation a) how successful the size exclusion has been i.e. is all the functional nanodisc complex in the second peak or is the lack of mono-dispersion indicative of significant loss of the complex in the larger peak or dissociated subunits in the smaller peaks? In addition, the question as to the ATPase activity in these additional peaks was not answered. The ATPase activity in each of the other 3 observed peaks would also go some way to answering the question as to the nature and composition of these peaks. Again, I accept that the homodimer OpuA has been determined in the peak 2 of the SEC (~9-10 Elution volume) previously (Sikkema et al 2021), but where additional peaks also observed in this publication? And if so have their composition been determined? I still think actually showing the SDS-PAGE referred to would be beneficial. Or, at least, providing an explanation of the peaks in the text with reference to the other publications and clarification as to what these peaks are. My apologies if what I was asking in this point was not entirely clear in my first submission comments but these are important questions.
- The same question for the previous point could also be asked of the second peak on Fig. 7A - what is it and was this determined?
Author Response
See second rebuttal, attached.
